# Determining the Pharmacokinetic Properties of Two Different Doses of Meloxicam in Barred Owls (*Strix varia*)

**DOI:** 10.3390/ani14213086

**Published:** 2024-10-26

**Authors:** Zoey N. Lex, Laken Russell, Corinne Mayer, Jordan Padlo, Kimberly L. Boykin, Mark G. Papich, Mark A. Mitchell

**Affiliations:** 1Department of Veterinary Clinical Sciences, School of Veterinary Medicine, Louisiana State University, Skip Bertman Drive, Baton Rouge, LA 70803, USA; zlex1@lsu.edu (Z.N.L.); lrussell2@lsu.edu (L.R.); ccmayer18@gmail.com (C.M.); jpaldo1@lsu.edu (J.P.);; 2Clinical Pharmacology Laboratory, College of Veterinary Medicine, North Carolina State University, Raleigh, NC 27607, USA; mgpapich@ncsu.edu

**Keywords:** Barred owls, *Strix varia*, avian, raptors, meloxicam, non-steroidal anti-inflammatory drugs, analgesic, pharmacokinetics

## Abstract

Wildlife are negatively affected by human activities, and these encounters often lead to traumatic injuries requiring treatment. Unfortunately, our understanding of analgesics for injured wildlife is limited. The barred owl (*Strix varia*) is the most common raptor presented to the Wildlife Hospital of Louisiana, and, to date, there are no studies evaluating analgesics for this species. The purpose of this study was to measure the pharmacokinetic properties of two different doses (1 and 2 mg/kg intramuscularly) of meloxicam in barred owls. Meloxicam was found to reach subtherapeutic concentrations after 2–4 h. These results suggest standard twice-daily treatments may be insufficient.

## 1. Introduction

Avian species are commonly admitted to wildlife hospitals and rehabilitation centers [1], with the majority presenting for traumatic injuries such as fractures, wounds, head trauma, and electrocution. Unfortunately, the injuries are usually caused by direct or indirect anthropogenic activities such as gunshot wounds and collisions into cars, buildings, windows, or power lines [1,2,3,4,5,6,7,8,9,10]. These injuries usually require treatment with analgesics and anti-inflammatory medications to control pain and inflammation [11,12,13]; however, there is limited evidence-based research available characterizing the pharmacokinetic or pharmacodynamic properties of these drugs for raptors.

Owls represent one group of raptors with a lack of evidence-based research on therapeutics. To date, only four studies have measured the pharmacokinetics of different analgesics in a species of owl, and all are limited to a single species (great horned owl, *Bubo virginianus*) [14,15,16,17]. The Wildlife Hospital of Louisiana (Baton Rouge, LA, USA) has had a 3.2-fold increase in barred owl (*Strix varia*) cases presented over the past eight years (2023, 134; 2015, 42), with individuals of this species being most commonly presented for traumatic injuries [3,5,8,9,10]. Unfortunately, there are no therapeutic studies available for barred owls to guide analgesic treatment for these injuries. To properly care for these animals, more evidence-based research on therapeutics (e.g., analgesics) is needed to provide these animals with “best care” practices.

Meloxicam is a COX-2 preferential inhibitor non-steroidal anti-inflammatory drug with analgesic, anti-inflammatory, and antipyretic properties [18]. It has become a commonly used analgesic in wildlife medicine due to its availability and a handful of studies demonstrating its relative safety, minimal side effects, and efficacy in controlling painful stimuli in birds [19,20,21,22,23,24]. Recommended doses in birds range from 0.1 mg/kg up to 20 mg/kg subcutaneously (SC), per os (PO), or intramuscularly (IM) q6h to q24h [12,25]. There have only been a few studies on the pharmacokinetics of meloxicam in raptors, including Cape Griffon vultures (*Gyps coprotheres*), red-tailed hawks (*Buteo jamaicensis*), and great horned owls [14,21]. Wide species variability in plasma concentrations and elimination rates of meloxicam were demonstrated in these birds, with red-tailed hawks having the shortest elimination half-life and great horned owls having the highest plasma concentrations compared to any other avian species [14]. These pharmacokinetic studies further demonstrate the importance of having species-specific studies, even within assumed closely related species, to accurately dose patients.

The purpose of this study was to determine the harmonic means, times to maximum concentration, and elimination half-lives for single intramuscular doses of 1 mg/kg and 2 mg/kg meloxicam in barred owls. We hypothesized that both doses would not cause any adverse effects and provide appropriate plasma concentrations for at least 12 h but that there would be significant differences in the pharmacokinetic properties between doses.

## 2. Materials and Methods

The study protocol (#21-014) was approved by the Institutional Animal Care and Use Committee at Louisiana State University (Baton Rouge, LA, USA). Twelve adult barred owls of unknown sex that presented to the Wildlife Hospital of Louisiana at Louisiana State University, School of Veterinary Medicine, were used in this study. Body weights ranged from 515–990 g. The birds were housed in rehabilitation mews (15.2 m × 3 m × 5.5 m) constructed of wood slats (2.54 × 5.08 cm) and had a limestone gravel substrate. Perches were constructed of treated lumber and covered with astroturf or rope; dead branches harvested from the woods were also used as perches when available. The owls were fed frozen/thawed mice equivalent to 10% of their body weight daily. During the study (12 h-period), the birds were housed in a wall-mounted stainless-steel cage (71 × 55 × 71 cm) with a newspaper and astroturf substrate and an astroturf-covered polyvinyl chloride perch. Because the owls were fed at night, they were not offered food during the 12-hour trial; however, they were offered their standard meal immediately after collecting their last blood sample. A physical exam and complete blood count were performed to assess the health of the birds. The following inclusion criteria had to be met to be included in this study: a body weight > 500 g; a packed cell volume (PCV) > 30%; a complete white blood cell count (WBC) < 25 × 10^3^ cells/mL; no apparent infection; and a 2-week washout of any medications given in hospital prior to the study.

Once admitted into the study, barred owls were randomly assigned into two groups for the study using a random number generator (random.org): group 1 (*n* = 6), 1 mg/kg meloxicam; group 2 (*n* = 6), 2 mg/kg meloxicam. Each bird was weighed the morning of the study and appropriately dosed with meloxicam (Ostilox, 5 mg/mL, VetOne, Boise, ID, USA) intramuscularly into the right pectoralis major. One individual manually restrained the bird’s head and talons while a second individual collected blood samples. A cotton ball soaked with 70% isopropyl alcohol was used to part the feathers over the venipuncture site (apterium) for disinfection and to visualize the jugular vein. Blood samples (0.5 mL) were collected from the right or left jugular vein using a 26-gauge needle fastened to a 3-mL syringe prior to meloxicam administration (baseline), and at 30 min, 45 min, 1 h, 2 h, 4 h, 6 h, and 12 h after administration. The total blood collected from each bird (4 mL) was <1% of their body weight. Blood samples were immediately placed in lithium heparin microtainers (Becton and Dickinson, Franklin Lakes, NJ, USA) and centrifuged at 13,528× *g* for 3 min (LW Scientific Combo, Lawrenceville, GA, USA). The plasma was immediately removed and stored at −80 °F until being analyzed. All samples were transported overnight on ice to the North Carolina State University College of Veterinary Medicine Clinical Pharmacology Lab (Raleigh, NC, USA) for analysis.

Plasma meloxicam concentrations were measured by high-pressure liquid chromatography (HPLC). The instruments included a quaternary solvent delivery system, an ultraviolet (UV) detector, and a sample injector (1200 Series Agilent Technologies, Wilmington, DE, USA). The peaks were monitored with the UV detector at a wavelength of 365 nm. A 4.6 × 150 mm reverse phase column was used to separate chromatographic peaks (Zorbax SB-C8 Column, 4.6 × 150 mm, 5 µm, Agilent Technologies, Wilmington, DE, USA) at a flow rate of 1 mL/min. The mobile phase was 60% acetonitrile and 40% distilled water with an acidic pH modifier added.

Calibration, quality control (QC), and incurred plasma samples were all prepared in an identical manner. The samples were extracted by pipetting the plasma sample into a microcentrifuge tube and adding acetonitrile in a 2:1 ratio (acetonitrile: plasma). After vertexing, the tubes were centrifuged at 25,000× *g* for 5 min. The supernatant was collected and dried. The dry residue was reconstituted with a mixture of 60/40 water/acetonitrile, vortexed, and loaded into a Whatman™ syringeless filter device, pore size 0.2 µm. Twenty μL of the sample was injected into the HPLC system. The chromatographic peaks were monitored and integrated with Agilent OpenLab ChemStation software, version 2.8 (Agilent Technologies, Wilmington, DE, USA).

Peak identification was confirmed by the lack of interfering peaks from endogenous compounds in blank plasma samples with the same retention times as meloxicam. Quality control and calibration standards were prepared by dissolving a pure analytical reference standard of meloxicam sodium in pure methanol to a concentration of 1 mg/mL. Additional dilutions were made with a water/acetonitrile mixture. Calibration curve and QC samples were prepared by fortifying blank (control) banked barred owl plasma with meloxicam to create seven nominal concentrations of meloxicam ranging from 0.05 to 10 μg/mL and a blank. Fresh calibration curves were prepared for each run and accepted if the R^2^ value was 0.99 or greater. The limit of quantification (LOQ) was 0.05 μg/mL based on the lowest concentration on a linear calibration curve that met our acceptance criteria. Five blood samples collected from barred owls not otherwise involved in the study were spiked with meloxicam (10 μg/mL), centrifuged, and stored for 2 h, and meloxicam concentration was measured to identify any interference from lithium heparin in the plasma separator tubes.

Pharmacokinetic analysis—Analysis of the concentration vs time plots and pharmacokinetic analysis were performed with with Phoenix^®^ software (WinNonlin, Certara^®^, St. Louis, MO, USA). Noncompartmental analysis was performed to obtain initial estimates. Final results were obtained with compartmental analysis using the Phoenix^®^ software. A one-compartment model with first-order input was selected based on best-fit analysis, examination of residual plots, and visual examination of the plots for both doses. Primary parameters calculated included absorption (K01) and elimination rates (K10) and volume of distribution per fraction absorbed (VD/F). Secondary parameters included the K01 and K10 half-lives (T½), area under the curve (AUC), peak concentration (C_MAX_), time to peak concentration (T_MAX_), and clearance per fraction absorbed (CL/F). VD/F and CLF were not reported in the tables because F is not known.

The sample size for this pilot study (meloxicam: *n* = 12; 1 mg/kg group, *n* = 6; 2 mg/kg group, *n* = 6) was based on the following a priori assumptions: an alpha =0.05, a power = 0.8, and differences in the expected mean and standard deviations of the peak concentration (Cmax) of 7.5 µg/mL and 3.5 µg/mL between the 1 mg/kg and 2 mg/kg doses, respectively. The Shapiro-Wilk test, skewness, kurtosis, and q-q plots were used to evaluate the distributions of the continuous data. Normally distributed data are reported by the mean, standard deviation (SD), and minimum-maximum (min-max) values, while non-normally distributed data are reported by the medians, 25–75 percentiles (%), and min-max values. Baseline PCV, TS, and WBC values between groups were compared using an independent samples t-test. Independent samples t-tests were also used to determine if the pharmacokinetic properties (AUC, C_MAX_, T_MAX_, T½, K01, K10) differed between the two doses. Levene’s test was used to assess variance within the data. SPSS 25.0 (IBM Statistics, Armonk, NY, USA) was used to analyze the data. A P < 0.05 was used to determine statistical significance.

A search of the electronic medical records system used in the WHL (Wildlife Rehabilitation MD; The Wildlife Neighbors Database Project, Middletown, CA, USA) was done to determine the number of barred owls presented to the WHL from 2020–2022. The records were reviewed to determine the number of barred owls treated with meloxicam, the duration of treatment, and the outcome associated with treatment. A chi-squared test was used to determine if treatment with meloxicam was associated with a significantly different outcome (released vs not released) from those birds not treated with meloxicam. Necropsies of barred owls presented during the same time (2020–2022) that received 2 mg/kg meloxicam treatments were also reviewed to screen the birds for histopathological lesions attributed to meloxicam toxicity.

## 3. Results

All birds recruited for the study were found to be in good general health based on their physical examination findings and hematologic values being within published reference intervals for the species (Table 1) [26,27]. Two owls did have underlying issues that prevented their release (e.g., permanent leg injury and retinal disease) but had normal mentation, appetite, and defecation/micturition events. There were no significant differences in the PCV (t = −0.449, *p* = 0.663), TS (t = −0.995, *p* = 0.377), or WBC counts (t = −1.2, *p* = 0.258) between the owls receiving the 1- and 2 mg/kg doses of meloxicam. No adverse clinical side effects were observed during the trial or the birds’ hospitalization following the single injections of either 1 mg/kg or 2 mg/kg dose of meloxicam. Of the 12 birds involved in the study, 10 were released, one was euthanized for a chronic injury that would not allow release, and one was kept as a resident bird for education programs due to a chronic visual deficit (Table 1).

Plasma concentration-time curves were generated for 4 and 6 of the barred owls receiving the 1 mg/kg and 2 mg/kg doses of meloxicam, respectively. Two owls could not be analyzed from the 1 mg/kg data because their initial high concentrations were considered outliers due to a potential issue of sample contamination at the later sampling periods. (Figure 1) The samples in question for barred owls #663 and #664 were 2.3–5.1 and 2.7–5.8; 2.5–25.9 and 2.7–27.2; and 17.9–107.7 and 14.7–88 times higher than samples for the other owls at the 4, 6, and 12 h sampling periods, respectively (Appendix A). A one-compartment model best described the pharmacokinetic properties for the 1 mg/kg and 2 mg/kg intramuscular doses of meloxicam in barred owls.

There were significant differences in the AUC (t = −2.8, *p* = 0.015), Cmax (t = −2.4, *p* = 0.017), absorption rate half-life (K01_1/2_) (t = −2.1, *p* = 0.032), and time to peak concentration (Tmax) (t = −2.5, *p* = 0.016) between the two dosing groups (Table 2). There were no significant differences in the absorption rate (K01; t = −1.6, *p* = 0.07), elimination rate (K10; t = −0.8, *p* = 0.215), or elimination rate half-life (K10_1/2_; t = −0.4, *p* = 0.3550) between groups. Barred owls given the 2 mg/kg dose had an AUC, Cmax, K10_1/2_, and Tmax that were 2.7,1.9, 4, and 2.2 times higher than the birds receiving the 1 mg/kg dose, respectively. Plasma concentrations were below the limits of quantification (0.1 µg/mL) by 12 h in most birds in both the 1 mg/kg and 2 mg/kg doses (Figure 1, Table 2).

From 2020–2022, 133 barred owls were presented to the LSU WHL. From this cohort, 46 (34.6%) were euthanized within 24 h, 9 (6.8%) died within 24 h, 12 (9%) died after 24 h, 17 (12.8%) were euthanized after 24 h, and 49 (36.8%) were released. Traumatic injuries were diagnosed in 88% (117/133) of the animals, representing the majority of cases. Other diagnoses included orphans (3%; 4/133), neurologic signs with no overt evidence of trauma (3%; 4/133), emaciation (1.5%; 2/133), and idiopathic disease (4.5%; 6/133). Seventy-six (57.1%, 76/133) barred owls from this cohort received treatment with 2 mg/kg of meloxicam for varying durations (1–32 days). Forty-nine (64.4%, 49/76) birds from this cohort were released, and 45 (91.8%, 45/49) received treatment with 2 mg/kg meloxicam PO BID; 4 (8.2%, 4/49) released barred owls did not receive any meloxicam. Of the 84 (63.2%, 84/133) barred owls that either died or were euthanized, 31 (36.9%, 31/84) received 2 mg/kg of meloxicam PO BID, and 53 (63.1%, 53/84) did not receive any meloxicam. There was a significantly increased likelihood of birds being released if they were dosed with 2 mg/kg meloxicam PO BID (Χ^2^ = 38.1, *p* < 0.00001). During this study period, 6 necropsies were performed on owls that died and had been treated with 2 mg/kg doses of meloxicam PO BID for (range of treatment days with meloxicam: 3–26). None of the birds were found to have gastrointestinal lesions associated with gastritis or ulceration on gross or histologic evaluation. Four of the birds had evidence of mild to moderate visceral gout, and 1 had lipid vacuoles in hepatocytes. The specific renal histopathology reported included the following lesions: mild mineral deposits within the tubules and focal area of lymphocytes within the interstitium; distension of renal tubules with degenerate epithelium and evidence of urate tophi; tubular dilation, degeneration, and necrosis associated with intratubular urate crystals; multifocal replacement and effacement of tubules by urate tophi; and necrosis and granulomatous lesions in the cortical parenchyma with myriad colonies of rod-shaped bacteria.

## 4. Discussion

The results of this study supported our first hypothesis that a single dose of meloxicam at 1- and 2 mg/kg intramuscularly would not be associated with any adverse effects. All 12 birds were found to be unaffected by the trial, and the majority were released after completing their treatments. The only bird that was euthanized following the study was attributed to an injury that would not allow release, and there was (unfortunately) no facility available to place the non-releasable barred owl. No histopathologic changes associated with meloxicam toxicity were identified in this bird. Further, the results of our retrospective cross-sectional study of the WHL barred owls found a significant association between release and birds being treated with 2 mg/kg meloxicam PO BID. The authors initially started treating barred owls at these doses based on the results of other species, where doses of 0.5–1 mg/kg were found to be inadequate [14]. It was not until later that the authors were able to pursue this pharmacokinetic study. Based on the results of the retrospective data, the birds tolerated the meloxicam well, as a majority were released, and, albeit limited to six animals, there were no obvious post-mortem lesions suggesting that meloxicam at that dosing schedule was associated with acute toxicity. While the authors did not evaluate the pharmacokinetics of 2 mg/kg PO meloxicam in the barred owls, based on the results in great-horned owls using IV and PO dosing [14], it is likely that the drug would have been similarly excreted and not available as desired for a twelve-hour dosing schedule. The authors used the 2 mg/kg PO BID dosing strategy because our retrospective results suggested we saw positive results based on the clinical responses of the barred owls, but it is possible that all the supportive care provided to the birds was the reason for the success. This should further reinforce why it is important to perform these pharmacokinetic studies, as well as pharmacodynamic studies that evaluate the birds’ responses to the drug. Moreover, studies assessing barred owls’ responses to pain are needed.

The authors’ second hypothesis was also partially proved because there were significant differences in the AUC, K01_1/2_, Cmax, and Tmax between the two doses of meloxicam in barred owls, with a 1.9–4 time increase in these parameters in the 2 mg/kg dosed birds compared to the 1 mg/kg birds; there were no significant differences in the K01, K10, or K10_1/2_ parameters between groups. The plasma concentrations of meloxicam achieved based on AUC and Cmax were proportional to the dose with approximately doubling of these values for the 2× higher dose. Therefore, when the dose of meloxicam was doubled, the concentration of meloxicam can be expected to double too, suggesting that there is no saturation of the absorption process. There was no evidence of dose-dependent pharmacokinetics with similar elimination half-life, clearance, and volume of distribution between doses. The second part of our first hypothesis was not accepted because absorption was fast, reaching a peak (Tmax) within an hour, regardless of the dose, and the elimination half-life was approximately one hour for both doses. This was similar to pharmacokinetics for meloxicam in other raptor species (great horned owl, red-tailed hawk), where a short elimination half-life and rapid Tmax were recorded following intravascular doses of 0.5 mg/kg (Table 3) [14,21]. Oral 0.5 mg/kg doses of meloxicam in great horned owls and red-tailed hawks had longer elimination ½-lives but had a much lower Cmax in comparison to the intravenous doses [14]. Based on the intramuscular doses used in this study, the authors believe this would also be true for the barred owls. This suggests that higher oral doses of meloxicam may be needed in these raptor species because of the lower bioavailability and lower blood concentrations compared to intravenous and intramuscular doses.

Even with the increasing Cmax and AUC measured for the 1- to 2 mg/kg doses, the elimination half-lives were only one hour for both doses and meloxicam plasma concentrations were <3.5 µg/mL after 1–2 h in the 1 mg/kg dosing group and between 2–4 h in the 2 mg/kg dosing group. This may suggest that increasing doses could increase the time and amount of meloxicam concentration in the blood but still makes it questionable if we could give safe and adequate dosing intervals in this species. However, it is recognized that plasma concentrations of NSAIDs do not predict therapeutic response. The tissue concentrations, which may persist much longer than in the plasma, are responsible for therapeutic effects [28]. In mammals (dogs, cats, horses, people), meloxicam is administered once daily regardless of the elimination half-life. Without more study, we cannot speculate on the clinical effects of incrementally increased doses or more frequent intervals of administration in birds. Adverse effects of meloxicam in mammals (dogs, cats) are dose-dependent, possibly because meloxicam COX-2 selectivity is lost at higher concentrations. It is undetermined if these dose-dependent effects occur in birds.

The results of this study confirmed that 1- and 2 mg/kg intramuscular doses of meloxicam will not provide appropriate plasma concentrations in barred owls for 12 h. Studies in other avian species have suggested that using these higher doses of 1–2 mg/kg of meloxicam provides quantifiable effects of analgesia and appears to be safe [22,23]. Through pharmacodynamic and pharmacokinetic studies of intramuscular meloxicam in Hispaniola parrots (*Amazona ventralis*), an effective plasma concentration of 3.5 ± 2.2 μg/mL was found to provide appropriate analgesia for experimentally induced arthritis [22,29]. In comparison, there was a fast increase in meloxicam plasma concentrations in the barred owls that subsequently dropped below 3.5 µg/mL after 1–2 h and 2–4 h in the 1 mg/kg dose and 2 mg/kg dosing groups, respectively. These results suggest that these doses of meloxicam in barred owls may have questionable efficacy for analgesia, at least for induced arthritic disease, and that the dosing frequency would likely need to be more than twice daily. However, as mentioned previously, dose frequency does not affect the efficacy of meloxicam in mammals because it is administered once daily regardless of the half-life. This occurs because it is the tissue concentrations, not the plasma concentrations, that determine the efficacy of the NSAIDs [30]. It is not known if these principles also apply to birds. Pharmacodynamic studies are ultimately needed to determine the analgesic properties of meloxicam for barred owls. Moreover, studies determining the plasma concentrations of meloxicam required for managing inflammation in these species are needed.

Meloxicam is generally considered a safe non-steroidal anti-inflammatory choice due to its COX-2 preferential inhibition, with the most common adverse effects being gastrointestinal distress and, less commonly, renal toxicity [18]. In avian species, studies using variable doses (1–2 mg/kg) of meloxicam given once or twice daily for up to 15 days did not cause clinical signs or lesions within the gastrointestinal or renal system [31,32,33,34]. In a study evaluating the potential adverse effects of serial 5 mg/kg oral doses of meloxicam in chickens (*Gallus gallus domesticus*), 64% (7/11) of birds developed significant lesions on histopathology, including acute renal tubular injury and gout [35]. In another study evaluating serial high oral doses of up to 20 mg/kg meloxicam in American kestrels (*Falco sparverius*), the authors concluded that higher doses did not result in nephrotoxicity but could have the potential to cause adverse gastrointestinal and hepatic effects. There was a significant correlation between the degree of hepatocellular vacuolization and increasing meloxicam dose, and, although not considered significant, 22% (2/9) of the birds receiving 20 mg/kg were observed to have gastric mucosal ulceration on necropsy [19]. Of the historic barred owl cases at the Wildlife Hospital of Louisiana that received serial 2 mg/kg doses of oral meloxicam and were necropsied, 67% (4/6) of the birds were found to have evidence of visceral or renal gout but no signs of gastrointestinal or hepatic lesions. It was not possible to determine whether the gout lesions were associated with other underlying disease processes these birds were presented for or if they were related to the treatment with meloxicam. Of these six birds, each was given subcutaneous fluids (maintenance rate plus deficit of ≥100 mL/kg/day) for 2–7 days during hospitalization; the decision to stop supplemental fluids was based on the birds eating without assistance. Five of these birds received 2 mg/kg meloxicam doses PO BID for 3–7 days, and the sixth bird received the same dose for 26 days. There were no adverse effects in any of the barred owls used in the pharmacokinetic study; however, these birds only received a single dose of meloxicam. Because only a single owl in the pharmacokinetic study was necropsied (chronic leg injury), the direct adverse effects of a single dose of 1- or 2 mg/kg meloxicam on the gastrointestinal tract, kidneys, and liver could not be fully evaluated. Future studies should assess the potential risks for hepatotoxicity, nephrotoxicity, and gastrointestinal adverse effects when using these doses and potentially higher doses of meloxicam in barred owls and other avian species.

There were several limitations associated with this study that should be addressed. The study had a limited sample size of six animals in each dosing group, and two birds had to be removed from the pharmacokinetic analysis in the 1 mg/kg data due to potential contamination. The samples for the two birds that were removed from the analysis were retested, and the results were similar. Because these samples were 2.3–107.7 times higher than the results obtained for the other four birds in the 1 mg/kg group and similarly higher than the concentrations achieved in the 2 mg/kg group, they were removed. It is possible that there was a biological reason for these differences, such as CYP450 enzyme deficiency relevant to meloxicam metabolism in owls; however, the birds were sampled from a similar population, and it might be expected that this difference would have also been observed in the other owls receiving 1 mg/kg meloxicam, as well as the birds in the 2 mg/kg dosing group. Unfortunately, elaborating further on these potential sources for the unexpectedly high concentrations was beyond the scope of this study but is worth investigating in the future. While the sample size was limited, the authors did find that differences did exist between the two doses (AUC, Cmax, K011/2, and Tmax.), confirming that widespread type II errors were not present. Due to limited resources, we did not perform serum chemistries to assess each bird’s liver (bile acids) and kidney function (calcium to phosphorus ratio, uric acid) prior to and after the study. This could have potentially impacted pharmacokinetics; however, we feel it was unlikely to impact our final results because all but two birds were healthy enough to quality for release following the conclusion of the study, and the two exceptions to this were unable to be released for reasons unrelated to the meloxicam administration. A crossover study would have also been beneficial for comparing the effects of each dose within barred owl to limit the variability of the results. This type of study design would have been preferred and may have helped to explain the two outliers in the 1 mg/kg dosing group. Ultimately, we decided against this because these birds were being rehabilitated for release, and we did not want to delay this process. Finally, the results from the cross-sectional study need to be interpreted with caution because there was no control over the types of presentations, and patients received treatments based on the individual clinician treating the case. A longitudinal cohort study would be a preferred type of study to better assess the potential causative link between treatment (2 mg/g PO BID) and release rates.

The results of this study and others continue to support that twice daily dosing of meloxicam at the currently recommended doses in raptors is most likely inadequate and that we would most likely need to increase the amount or frequency of dosing [14,21]. This would require future studies to evaluate the pharmacokinetics and safety of these higher doses through different routes of administration. Other studies looking at different formulations of meloxicam, such as the more recently compounded, sustained-release meloxicam (ZooPharm, Windsor, CO, USA), may be beneficial, but initial studies in avian species show questionable differences from the regular formulation [36,37]. Concurrent pharmacodynamic studies would also be needed to further elucidate the clinical efficacy of these doses in this species.

## 5. Conclusions

Meloxicam given intramuscularly to barred owls at 1- and 2 mg/kg doses was shown to have a rapid elimination half-life and reach plasma concentrations considered to be therapeutic in another avian species for only two to four hours. These results suggest that the currently recommended dosing of 1–2 mg/kg meloxicam twice daily may be unlikely to maintain the plasma concentrations anticipated to be therapeutic, and practical dosing options are questionable for this non-steroidal anti-inflammatory medication in this raptor species. Additional studies evaluating the pharmacokinetics and pharmacodynamics of more frequent and possibly higher doses to assess the safety of meloxicam in barred owls are needed.

## Figures and Tables

**Figure 1 animals-14-03086-f001:**
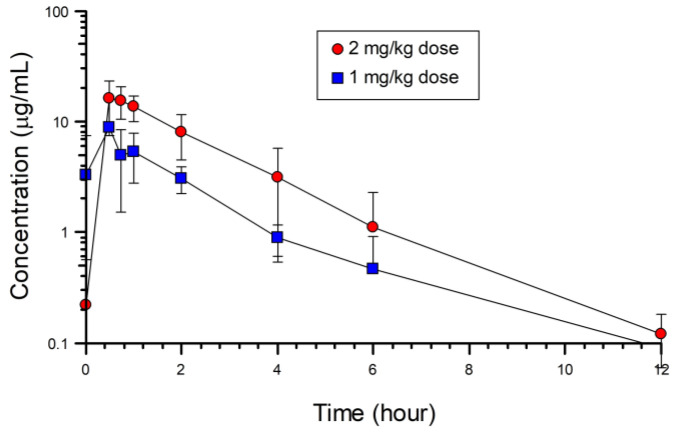
Linear plot of plasma concentrations (µg/mL ± SD) over time (0–12 h) after single intramuscular injection of 1 mg/kg (*n* = 5) or 2 mg/kg (*n* = 6) meloxicam in barred owls.

**Table 1 animals-14-03086-t001:** Hematologic values and outcomes of barred owls given meloxicam at 1 mg/kg (*n* = 6) or 2 mg/kg (*n* = 6) intramuscularly.

Barred Owl	Meloxicam Dose	PCV (%)	TS (g/dL)	WBC (cells/µL)	Outcome
21–615	1 mg/kg	36	5.1	19,000	Released
21–126	1 mg/kg	33	4.9	25,000	Released
21–664	1 mg/kg	44	5.4	16,000	Released
21–663	1 mg/kg	38	4.6	8500	Released
21–747	1 mg/kg	45	5	19,000	Released
21–968	1 mg/kg	34	6.1	7000	Released
21–273	2 mg/kg	39	5.8	23,000	Euthanized; chronic leg injury
21–571	2 mg/kg	36	6.1	21,000	Released
21–529	2 mg/kg	47	5.7	18,000	Resident; Chronic visual deficit
21–495	2 mg/kg	37	5.6	20,000	Released
21–771	2 mg/kg	34	4.1	16,000	Released
21–980	2 mg/kg	45	5.8	18,000	Released

**Table 2 animals-14-03086-t002:** Pharmacokinetic properties of meloxicam in barred owls after a single intramuscular dose of 1- or 2 mg/kg. Significant differences were noted between doses for the AUC, Cmax, K01_1/2_, and Tmax.

Parameter	Dose	Mean	SD	Min–Max
AUC	1 mg/kg 2 mg/kg	17.14 ^a^ 39.5 ^a^	5.2 18.5	12.9–26 20.8-60.7
Cmax	1 mg/kg 2 mg/kg	8.8 ^b^ 17.1 ^b^	1.9 7.2	5.7–11 9.4-30.4
K01	1 mg/kg 2 mg/kg	13.7 6.5	5.4 8.3	6–20.5 1.2–23.3
K01_1/2_	1 mg/kg 2 mg/kg	0.06 ^c^ 0.24 ^c^	0.03 0.18	0.03–0.12 0.03–0.57
K10	1 mg/kg 2 mg/kg	0.6 0.75	0.13 0.32	0.47–0.78 0.4–1.22
K10_1/2_	1 mg/kg 2 mg/kg	1.15 1.07	0.24 0.43	0.89–1.47 0.57–1.74
Tmax	1 mg/kg 2 mg/kg	0.26 ^d^ 0.57 ^d^	0.08 0.26	0.18–0.39 0.16–0.83

^a^ *p* = 0.015; ^b^
*p* = 0.017; ^c^
*p* = 0.032; ^d^
*p* = 0.016. Legend-AUC, area-under-the-curve; Cmax, peak concentration; K01, absorption rate and corresponding half-life; K10, elimination rate and corresponding half-life; and Tmax, time to peak concentration.

**Table 3 animals-14-03086-t003:** Pharmacokinetic properties of 1-and 2-mg/kg intramuscular doses of meloxicam in barred owls from the current study and 0.5 mg/kg oral and intravenous doses of meloxicam in red-tailed hawks and great horned owls [14].

	RTHA PO (0.5 mg/kg)	RTHA IV (0.5 mg/kg)	GHOW PO (0.5 mg/kg)	GHOW IV (0.5 mg/kg)	BDOW IM (1 mg/kg)	BDOW IM (2 mg/kg)
Cmax (µg/mL)	0.182	0.53	0.368	3.77	9.05	17.13
Tmax (h)	0.73	0.27	7.8	0.25	0.26	0.57
AUC (h × µg/mL)	0.462	0.544	3.23	4.17	15.6	39.55
Elimination ½ life (h)	3.97	0.49	5.07	0.78	0.99	1.07
Clearance (mL/h/kg)	543	1675	175	154	64.5	61.7

Abbreviations: AUC, area-under-the-curve; Cmax, peak concentration; and Tmax, time to peak concentration.

## Data Availability

The individual study subject data can be acquired by contacting the corresponding author.

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
