# Peer review of "Determining the Pharmacokinetic Properties of Two Different Doses of Meloxicam in Barred Owls (Strix varia)"

_animals, 2024, doi:10.3390/ani14213086_

Round 1
Reviewer 1 Report
Comments and Suggestions for Authors
The authors deal with the pharmacokinetic of meloxicam after intramuscular injection at dosages of 1 and 2 mg/kg body weight in barred owls. Overall, the manuscript is well written and objectives. The experimental design and analysis have no major flaws, and meet standard criteria. The data from this study is informative and useful for veterinary practice. However, there are a few points not clear for this reviewer. Materials and Methods
- The volume of blood collection per each time should be addressed.
- The reference and procedures for sample preparation should be clarified.
- For the sample preparation, Are there any internal standard used for analysis?
- L91: the abbreviation of packed cell volume, complete white blood counts should be added.
- L125: The unit of “um” should be amended.
- L157: “Cmax of 7.5 mg/ml and 3.5 mg/ml between the 1 mg/kg and 2 mg/kg doses, respectively” This sentence should be confirmed. It is not clear for this reviewer.
- L196: What does the “contamination of samples” refer to?
Results
- For pharmacokinetic evaluation, how was AUC calculated? Together with this, it perhaps necessary to calculate the AUC including, AUC0-t and AUC0-inf.
- The method validation parameters should be clarified.
- L232: What is the exact meaning of “(range 3-26 days)”?
- Figure 1: The linear and logarithmic plots were seen. However, I was wondering that one figure should be enough for addressing in the manuscript to avoid repetition.
Comments on the Quality of English Language
English language is good and easy to understand.
Author Response
Reviewer 1
The authors deal with the pharmacokinetic of meloxicam after intramuscular injection at dosages of 1 and 2 mg/kg body weight in barred owls. Overall, the manuscript is well written and objective. The experimental design and analysis have no major flaws and meet standard criteria. The data from this study is informative and useful for veterinary practice. However, there are a few points not clear for this reviewer.
Authors: Thank you for your kind words and commitment to reviewing our manuscript.
Materials and Methods
- The volume of blood collection per each time should be addressed.
Authors: Thank you. We addressed this in lines 101-105. We specifically noted that we collected 0.5mL samples and a total of 4mL : “Blood samples (0.5 mL) were collected from the right or left jugular vein using a 26-gauge needle fastened to a 3-mL syringe prior to meloxicam administration (baseline), and at 30 minutes, 45 minutes, 1 hour, 2 hours, 4 hours, 6 hours, and 12 hours after administration. The total blood collected from each bird (4mL) was <1% of their body weight.”
Was there another way you wanted this reported?
Reviewer 1: The reference and procedures for sample preparation should be clarified.
Author: We are sorry but we are unsure of your specific question. In lines 105-111 we detailed the methods of how we handled the initial blood samples for preparation and storage:
Blood samples were immediately placed in lithium heparin microtainers (Becton and Dickinson, Franklin Lakes, NJ, USA) and centrifuged at 13,528 g for 3 minutes (LW Scientific Combo, Lawrenceville, GA, USA). The plasma was immediately removed and stored at -80oF until being analyzed. All samples were transported overnight on ice to the North Carolina State University College of Veterinary Medicine Clinical Pharmacology Lab (Raleigh, NC, USA) for analysis.
And for lines 112- 142 we outlined the procedures in detail for the pharmacokinetic analysis:
Plasma meloxicam concentrations were measured by high-pressure liquid chromatography (HPLC). The instruments included a quaternary solvent delivery system, ultraviolet (UV) detector, and sample injector (1200 Series Agilent Technologies, Wilmington DE, USA). The peaks were monitored with the UV detector at a wavelength of 365 nm. A 4.6 x 150 mm reverse phase column was used to separate chromatographic peaks (Zorbax SB-C8 Column, 4.6 x 150 mm, 5 µm, Agilent Technologies) at a flow rate of 1 mL/min. The mobile phase was 60% acetonitrile and 40% distilled water with an acidic pH modifier added.
Calibration, quality control (QC), and incurred plasma samples were all prepared in an identical manner. The samples were extracted by pipetting the plasma sample into a microcentrifuge tube and adding acetonitrile in a 2:1 ratio (acetonitrile: plasma). After vertexing, the tubes were centrifuged at 25,000 g for 5 minutes. The supernatant was collected and dried. The dry residue was reconstituted with a mixture of 60/40 water/acetonitrile, vortexed and loaded into a Whatman™ syringeless filter device, pore size 0.2 µm. Twenty μL of the sample was injected into the HPLC system. The chromatographic peaks were monitored and integrated with Agilent OpenLab ChemStation software (Agilent Technologies).
Peak identification was confirmed by the lack of interfering peaks from endogenous compounds in blank plasma samples with the same retention times as meloxicam. Quality control and calibration standards were prepared by dissolving a pure analytical reference standard of meloxicam sodium in pure methanol to a concentration of 1 mg/mL. Additional dilutions were made with a water/acetonitrile mixture. Calibration curve and QC samples were prepared by fortifying blank (control) banked barred owl plasma with meloxicam to create seven nominal concentrations of meloxicam ranging from 0.05 to 10 μg/mL, and a blank. Fresh calibration curves were prepared for each run and accepted if the R2 value was 0.99 or greater. The limit of quantification (LOQ) was 0.05 μg/mL based on the lowest concentration on a linear calibration curve that met our acceptance criteria. Five blood samples collected from barred owls not otherwise involved in the study were spiked with meloxicam (10 μg/mL), centrifuged, and stored for 2 hours, and meloxicam concentration was measured to identify any interference from lithium heparin in the plasma separator tubes.
What else would the reviewer like to see? For the reference, we used an analytical reference standard purchased from USP (www.USP.org) and prepared according to the USP standards.
- For the sample preparation, Are there any internal standard used for analysis?
Authors: No, we used an external standard method for calibration. An internal standard method was not needed.
- L91: the abbreviation of packed cell volume, complete white blood counts should be added.
Authors: Added. Thank you! Sorry we missed that.
- L125: The unit of “um” should be amended.
Authors: Great catch, thanks! Added : µm
- L157: “Cmax of 7.5 mg/ml and 3.5 mg/ml between the 1 mg/kg and 2 mg/kg doses, respectively” This sentence should be confirmed. It is not clear for this reviewer.
Authors: This is standard method for estimating sample size for studies a priori. Comparisons for continuous data (ie, Cmax) require a priori estimates for the mean difference between the groups and their estimated standard deviations. We used experience and published data to help identify these numbers. Not sure how else to explain them, sorry.
- L196: What does the “contamination of samples” refer to?
Authors: We don't know what caused these unusually high concentrations in these samples. It was possibly due to some sort of contamination at the time of sample processing. Because they were extreme outliers, we excluded these from the analysis.
- For pharmacokinetic evaluation, how was AUC calculated? Together with this, it perhaps necessary to calculate the AUC including, AUC0-t and AUC0-inf.
Authors: As it stated in the paper, we performed a noncompartmental analysis but did not show these results. (It just clutters up the results section - line 146.) We used these values for initial estimates. Then we proceeded with a compartmental analysis as described starting on line 146-153. Obviously when performing a compartmental analysis, the total AUC is calculated by integration. There is no such thing as AUC 0-T for a compartmental analysis. All results are for zero to infinity.
- The method validation parameters should be clarified.
Authors: We do not know what else the reviewer is asking for here. The method was based on prior studies. We published the method in prior papers from our laboratory. We already stated the validation steps starting on line 136 - 141 that describes our calibration curve acceptance criteria, and determination of the limits of quantification.
- L232: What is the exact meaning of “(range 3-26 days)”?
Authors: That is the range of days for meloxicam treatment at 2 mg/kg PO BID. We added “(range of treatment days with meloxicam: 3 – 26)“ to further clarify. Thanks!
- Figure 1: The linear and logarithmic plots were seen. However, I was wondering that one figure should be enough for addressing in the manuscript to avoid repetition.
We removed one figure to limit any confusion.
Reviewer 2 Report
Comments and Suggestions for Authors
Line 71-72: “We hypothesized that both doses would be safe and provide appropriate plasma concentrations for at least 12 hours…” This study was not adequately powered to assess safety. With 6 owls, the 95% CI that treatment is not associated with a significant adverse event is between 0-50%, assuming no adverse event occurred in the study. For accuracy, the investigators can hypothesize that no adverse events would be observed or just omit this phrase referring to safety.
Line 163-164: “Independent samples t-tests were also used to determine if the pharmacokinetic properties (AUC, CMAX, TMAX, T½, K01, K10) differed between the two doses.” Is the author sure that all of these PK values were normally-distributed (and thus suitable for comparisons using t-tests for which assume normal distributions)? A quick examination of Table 2 suggests that this is likely not the case for 2mg/kg K01 (high max values relative to mean may be due to tailing or outliers), and possibly other values. (In the reviewer’s experience, PK parameters from small samples frequently deviate from normality; it would be highly usual for error terms from every model parameter to not deviate from normality.)
Line 172-174: “A chi-squared test was used to determine if treatment with meloxicam was associated with a significantly different outcome from those birds not treated with meloxicam.” It is not clear what the investigators mean by a different outcome: better analgesia, increased feed intake or weight gain, reduced hematocrit, AKI, etc? It is also not clear what was routinely monitored/measured with respect to looking for a different outcome. (AKI will be less frequently diagnosed if chemistry panels or urine tubular enzyme measurements are not performed, in which case comparisons between treated/untreated animals may be less meaningful.)
Line 194-196: “Two owls could not be analyzed from the 1mg/kg data because their initial high concentrations were considered outliers due to a potential issue of contamination of samples.” More information is needed regarding the sample contamination and censoring of one-third of the data for one of the meloxicam doses. How were the samples contaminated, and how was this contamination prevented in samples from the other owls? Are the investigators sure that high peak concentrations observed in 2 owls was not due to inter-individual variability, and if so, how?
Line 241-242: “None of these lesions were specifically reported to be a factor from receiving meloxicam but cannot be ruled out.” This is an inferential statement appropriate for the discussion, but not results. Furthermore, from the authors’ introduction in this manuscript, there is likely not enough known about meloxicam renal toxicity in owls for there to be specific pathologic reports from which one can draw any conclusions as to whether meloxicam administration causative.
Line 244-245: “The results of this study supported first hypothesis, that a single dose of meloxicam at 1- and 2 mg/kg intramuscularly would be safe.” This statement is not true. The study was not adequately powered to answer questions about safety. The authors may conclude that meloxicam did not produce evidence of GI bleeding in the owls. The relationship between the renal tubular changes may be unrelated to the meloxicam, but a different study design would be needed to address this.
Line 250-252: “Further, the results of our retrospective cross-sectional study of the 250 WHL barred owls found that birds treated with 2 mg/kg meloxicam PO BID were significantly more likely to be released than untreated birds.” The authors imply a causative relationship between dose and outcome that may (or may not) be warranted. There is an association, but this should not imply causation between dose and outcome. For example, the analysis was not controlled for the severity of the disease. Wouldn’t a plausible explanation be that less sick owls received higher meloxicam doses (because clinicians may be more wary of anemia, GI or renal side-effects in sick animals), and the higher rates of release were really due to “high dose” owls being less sick? This is a critical reasoning/interpretation error that should be addressed/fixed in the discussion section.
Author Response
Reviewer 2
Line 71-72: “We hypothesized that both doses would be safe and provide appropriate plasma concentrations for at least 12 hours…” This study was not adequately powered to assess safety. With 6 owls, the 95% CI that treatment is not associated with a significant adverse event is between 0-50%, assuming no adverse event occurred in the study. For accuracy, the investigators can hypothesize that no adverse events would be observed or just omit this phrase referring to safety.
We were looking at safety from a clinical perspective for the experimental trial and that the animals would have no adverse issues (your point). The fact that all the birds had no complications post-trial and all survived for an extended period (one euthanized because not releasable) was what we considered “safe”. Thus, we have removed “be safe” because of your excellent point. It now reads: “We hypothesized that both doses would not cause any adverse effects and provide appropriate plasma concentrations for at least 12 hours, but that there would be significant differences in the pharmacokinetic properties between doses.”
Line 163-164: “Independent samples t-tests were also used to determine if the pharmacokinetic properties (AUC, CMAX, TMAX, T½, K01, K10) differed between the two doses.” Is the author sure that all of these PK values were normally-distributed (and thus suitable for comparisons using t-tests for which assume normal distributions)? A quick examination of Table 2 suggests that this is likely not the case for 2mg/kg K01 (high max values relative to mean may be due to tailing or outliers), and possibly other values. (In the reviewer’s experience, PK parameters from small samples frequently deviate from normality; it would be highly usual for error terms from every model parameter to not deviate from normality.)
Yes, please note we described the multiple methods we used to assess the distributions of the continuous data: Lines 158-159: “The Shapiro-Wilk test, skewness, kurtosis, and q-q plots were used to evaluate the distributions of the continuous data.” The reviewer is correct that small sample size and outliers can impact these results and thus why looking at multiple parameters are often necessary to not underestimate or overestimate the likelihood of misclassifying the distribution of the data. The corresponding author has used these methods for more than 25 years and teaches them in biostatistics. If the reviewer prefers us to publish the results using Mann Whitney U tests because they prefer a more conservative approach (statisticians like clinicians can be conservative or more liberal in approach based on preference) we can publish those. They were re-run that way and change the p values minimally (from table legend line 205):
current t-test: ap=0.015; bp=0.017; cp=0.032; dp=0.016.
MW U: ap=0.017; bp=0.009; cp=0.026; dp=0.041.
Line 172-174: “A chi-squared test was used to determine if treatment with meloxicam was associated with a significantly different outcome from those birds not treated with meloxicam.” It is not clear what the investigators mean by a different outcome: better analgesia, increased feed intake or weight gain, reduced hematocrit, AKI, etc? It is also not clear what was routinely monitored/measured with respect to looking for a different outcome. (AKI will be less frequently diagnosed if chemistry panels or urine tubular enzyme measurements are not performed, in which case comparisons between treated/untreated animals may be less meaningful.)
That is a great point. We did not clarify what “outcome’ was in that sentence. In lines 229-230 we note: “There was a significantly increased likelihood of birds being released if they were dosed with 2mg/kg meloxicam PO BID (Χ2=38.1, p <0.00001).”. Outcome is being released vs not released. We should have added that earlier. We have now added “outcome (released vs not released) to line 174. We have also addressed your later concern with lines 229-230 below.
Line 194-196: “Two owls could not be analyzed from the 1mg/kg data because their initial high concentrations were considered outliers due to a potential issue of contamination of samples.” More information is needed regarding the sample contamination and censoring of one-third of the data for one of the meloxicam doses. How were the samples contaminated, and how was this contamination prevented in samples from the other owls? Are the investigators sure that high peak concentrations observed in 2 owls was not due to inter-individual variability, and if so, how?
We don't know what caused these unusually high concentrations in these samples. It was possibly due to some sort of contamination at the time of sample processing with an interfering compound. Without additional undue examination of the samples using LCMS, we cannot determine the source of this contamination. Because they were extreme outliers, we excluded these from the analysis.
Line 241-242: “None of these lesions were specifically reported to be a factor from receiving meloxicam but cannot be ruled out.” This is an inferential statement appropriate for the discussion, but not results. Furthermore, from the authors’ introduction in this manuscript, there is likely not enough known about meloxicam renal toxicity in owls for specific pathologic reports from which one can draw any conclusions as to whether meloxicam administration causative.
This sentence has been removed. Thank you.
Line 244-245: “The results of this study supported first hypothesis, that a single dose of meloxicam at 1- and 2 mg/kg intramuscularly would be safe.” This statement is not true. The study was not adequately powered to answer questions about safety. The authors may conclude that meloxicam did not produce evidence of GI bleeding in the owls. The relationship between the renal tubular changes may be unrelated to the meloxicam, but a different study design would be needed to address this.
Please see our comments from above. Based on the changes there, we have made similar changes here: “ The results of this study supported our first hypothesis, that a single dose of meloxicam at 1- and 2 mg/kg intramuscularly would not be associated with any adverse effects.”
Line 250-252: “Further, the results of our retrospective cross-sectional study of the 250 WHL barred owls found that birds treated with 2 mg/kg meloxicam PO BID were significantly more likely to be released than untreated birds.” The authors imply a causative relationship between dose and outcome that may (or may not) be warranted. There is an association, but this should not imply causation between dose and outcome. For example, the analysis was not controlled for the severity of the disease. Wouldn’t a plausible explanation be that fewer sick owls received higher meloxicam doses (because clinicians may be more wary of anemia, GI or renal side-effects in sick animals), and the higher rates of release were really due to “high dose” owls being less sick? This is a critical reasoning/interpretation error that should be addressed/fixed in the discussion section.
Thank you. We agree. However, we did not see the statement as showing causation. We simply stated that the results found that birds that were treated with 2 mg/kg were more likely to be released—which they were. We never said it was because of the treatment, just that those birds were released at a higher rate. We certainly understand your point and did mean to include comments on the limitation associated with such a comparison on a historic set of data but left it out as an oversight. Thus, we have adjusted the sentences to the following:
Lines 252-254: Further, the results of our retrospective cross-sectional study of the WHL barred owls found a significant association between release and birds being treated with 2 mg/kg meloxicam PO BID.
Lines 379-383: “Finally, the results from the cross-sectional study need to be interpreted with caution because there was no control over the types of presentations and patients received treatments based on the individual clinician treating the case. A longitudinal cohort study would be a preferred type of study to better assess the potential causative link between treatment (2 mg/g PO BID) and release rates.”
Round 2
Reviewer 2 Report
Comments and Suggestions for Authors
Thank you for your revision. All comments have been satisfactorily addressed except for one. In response to censoring one-third of the 1mg/kg data, the investigators state, “We don't know what caused these unusually high concentrations in these samples. It was possibly due to some sort of contamination at the time of sample processing with an interfering compound. Without additional undue examination of the samples using LCMS, we cannot determine the source of this contamination. Because they were extreme outliers, we excluded these from the analysis.” It is possible that the samples were somehow contaminated. However, might other biological explanations be possible? Could these owls have been very calm with much lower cardiac output, leading to a much higher Cmax? Could some or all of the meloxicam have been injected intravascularly instead of into muscle tissue, resulting in a much higher Cmax? If the much higher concentrations occurred in later distribution/elimination phases, could differences be due to altered clearance from a CYP450 enzyme deficiency relevant to meloxicam metabolism in owls? There are a large number of possible explanations that remain unresolved because the reader has no access to the censored data (again, which comprises a significant fraction of the total data) and the magnitude and timing of the outlier values. This should be fixed by including a summary of the data that was censored from the analysis, either in the main manuscript or as a supplementary table (whichever the authors and editor prefer).
Author Response
Thank you for your revision. All comments have been satisfactorily addressed except for one. In response to censoring one-third of the 1mg/kg data, the investigators state, “We don't know what caused these unusually high concentrations in these samples. It was possibly due to some sort of contamination at the time of sample processing with an interfering compound. Without additional undue examination of the samples using LCMS, we cannot determine the source of this contamination. Because they were extreme outliers, we excluded these from the analysis.”
It is possible that the samples were somehow contaminated?
Yes, this is possible, but as we noted previously (unfortunately) we have no way of determining this. The samples of concern for barred owls 663 and 664 were re-analyzed with similar results (MP). We have attached a table to add as a supplementary per your request.
However, might other biological explanations be possible? Could these owls have been very calm with much lower cardiac output, leading to a much higher Cmax?
We don’t think this is likely because these are all wild owls held under the same conditions and heart rates were similar on examination (180-220 beats per minute). We are not sure how a lower cardiac output (which is impossible to measure) can affect Cmax (MP).
Could some or all of the meloxicam have been injected intravascularly instead of into muscle tissue, resulting in a much higher Cmax?
Again, that seems unlikely. Cmax was not much different in these birds compared to the others. But there were persistently high concentrations at the end (MP).
If the much higher concentrations occurred in later distribution/elimination phases, could differences be due to altered clearance from a CYP450 enzyme deficiency relevant to meloxicam metabolism in owls?
That is possible. Unfortunately, we did not have the resources to measure individual CYP450 enzyme activity in these birds. That was beyond the scope of our study. However, we did add a comment regarding this (please see at end of response)
There are a large number of possible explanations that remain unresolved because the reader has no access to the censored data (again, which comprises a significant fraction of the total data) and the magnitude and timing of the outlier values. This should be fixed by including a summary of the data that was censored from the analysis, either in the main manuscript or as a supplementary table (whichever the authors and editor prefer).
Data provided as a supplementary table for your review per your request.
Additionally, we have added the following to the results and limitations paragraph in the discussion to address the excellent suggestions you have made. We hope these are satisfactory. Thank you again for your assistance in improving this manuscript
Results
Lines195-200: Two owls could not be analyzed from the 1mg/kg data because their initial high concentrations were considered outliers due to a potential issue of sample contamination at the later sampling periods of samples. [Fig 1] The samples in question for barred owls #663 and #664 were 2.3-5.1 and 2.7-5.8; 2.5-25.9 and 2.7-27.2; and 17.9-107.7 and 14.7-88 times higher than samples for the other owls at the 4-, 6-, and 12-hour sampling periods, respectively.
Discussion
Lines 369-379 : The samples for the two birds that were removed from the analysis were retested but the results were similar. Because these samples were 2.3-107.7 times higher than the results obtained for the other four birds in the 1 mg/kg group, and similarly higher than the concentrations achieved in the 2 mg/kg group, they were removed. It is possible that there was a biological reason for these differences, such as CYP450 enzyme deficiency relevant to meloxicam metabolism in owls, however, the birds were sampled from a similar population and it might be expected that this difference would have also been observed in the other owls receiving 1 mg/kg meloxicam, as well as the birds in the 2 mg/kg dosing group. Unfortunately, elaborating further on these potential sources for the unexpectedly high concentrations was beyond the scope of this study but is worth investigating in the future.